# When in Need of an ESCRT: The Nature of Virus Assembly Sites Suggests Mechanistic Parallels between Nuclear Virus Egress and Retroviral Budding

**DOI:** 10.3390/v13061138

**Published:** 2021-06-13

**Authors:** Kevin M. Rose

**Affiliations:** Department of Molecular and Cell Biology, California Institute for Quantitative Biosciences, University of California—Berkeley, Berkeley, CA 94720, USA; kevin_rose@berkeley.edu

**Keywords:** enveloped virus budding, nuclear egress, ESCRT, membrane scission

## Abstract

The proper assembly and dissemination of progeny virions is a fundamental step in virus replication. As a whole, viruses have evolved a myriad of strategies to exploit cellular compartments and mechanisms to ensure a successful round of infection. For enveloped viruses such as retroviruses and herpesviruses, acquisition and incorporation of cellular membrane is an essential process during the formation of infectious viral particles. To do this, these viruses have evolved to hijack the host Endosomal Sorting Complexes Required for Transport (ESCRT-I, -II, and -III) to coordinate the sculpting of cellular membrane at virus assembly and dissemination sites, in seemingly different, yet fundamentally similar ways. For instance, at the plasma membrane, ESCRT-I recruitment is essential for HIV-1 assembly and budding, while it is dispensable for the release of HSV-1. Further, HSV-1 was shown to recruit ESCRT-III for nuclear particle assembly and egress, a process not used by retroviruses during replication. Although the cooption of ESCRTs occurs in two separate subcellular compartments and at two distinct steps for these viral lifecycles, the role fulfilled by ESCRTs at these sites appears to be conserved. This review discusses recent findings that shed some light on the potential parallels between retroviral budding and nuclear egress and proposes a model where HSV-1 nuclear egress may occur through an ESCRT-dependent mechanism.

## 1. Introduction

### ESCRTs Are Recruited to Sites of Viral Replication on Membranes

One of the most important trafficking pathways in the cell is the sorting of cargo once it has been internalized at the plasma membrane [1]. Cargo destined for degradation is typically modified with a small protein tag known as ubiquitin, a process referred to as ubiquitylation, and is compartmentalized within an internal vesicle where it will become part of the multivesicular body (MVB) on its way to the late endosome [2,3]. To ensure cargo has been compartmentalized during MVB biogenesis, the endosomal sorting complex required for transport (ESCRT) is recruited to the cargo by virtue of the ubiquitylation signal and directs cargo sorting [2,4]. The ubiquitin moiety from the cargo is detected by the ESCRT-I component TSG101 as well as the ESCRT-associated protein ALIX [5,6]. Once the cargo has been recognized, ESCRT-I can direct the sealing of the membrane compartment through sequential interactions with ESCRT-II and ESCRT-III, while ALIX can recruit ESCRT-III directly (Figure 1) [7]. ESCRT-III recruitment is temporally and spatially dynamic within the cell where it forms filaments that mediate the scission and sealing of a multitude of cellular membranes ranging from the nuclear envelope to the plasma membrane and many membranes in between [8,9,10,11,12]. Typically responsible for orchestrating the early events of membrane trafficking and scission in the cell, TSG101 and ALIX are commonly hijacked by viruses to facilitate viral replication and dissemination, one of the most well studied of cases coming from retroviruses like the Human Immunodeficiency Virus type 1 (HIV-1) [5,13]. In the case of HIV-1, the retroviral Gag protein encodes two late budding domains (L-domains). These short peptide motifs independently recruit ALIX (via a LYPXnL motif) and ESCRT-I (via a PTAP motif) to sites of viral budding and assembly by mimicking the peptide signal found within the cognate cellular partners of ALIX and TSG101 [5,13,14,15] (For review see [16]). Additionally, retroviral Gag is ubiquitylated by the ubiquitin ligase NEDD4, and ubiquitin modification is essential for ESCRT-mediated budding [17]. Recruitment of ESCRTs therein ensures genome packaging and proper maturation of the newly assembling retrovirus particle [18] (Figure 2a).

Despite the conservation of L-Domains seen in HIV-1 and host factors, there are notable exceptions to the consensus motifs commonly associated with ESCRT recruitment, namely in Simian Immunodeficiency Viruses (SIVs), or HIV-1-progenitor viruses, that encode divergent ALIX recruitment domains as determined by X-ray crystallography [19]. SIVs encode surprising species-dependent sequence variations in their ALIX recruitment domains with virus isolated from rhesus macaque (SIVmac) requiring a PYKEVTEDL motif, isolated from African green monkey (SIVagm) relying on a AYDPARKLL motif, and virus from sooty mangabey (SIVsmm) harboring a PYKEVTEDLLHLNSLF sequence for ALIX recruitment, each uniquely different from the consensus motif [20,21]. ALIX interactions with LYPXnL late domains are mediated by the central V domain of ALIX that has a hydrophobic cavity capable of accepting and stabilizing hydrophobic residues from an LYPXnL-containing binding partner [21]. ALIX recruitment by HIV-1 and SIVs with atypical motifs can be accomplished so long as the viruses encode a phenylalanine or tyrosine residue, with the proper flexibility to embed within the V domain of ALIX, independently of the immediate amino acid residues flanking the canonical LYPXnL motif [21,22].

Flaviviruses, another class of RNA virus that recruit ESCRT proteins, are classified on the basis of their tripartite use of a single stranded RNA genome as an mRNA, a genomic copy, and as a parental strand for transcription [23]. This is achieved through the encoding of a viral RNA-dependent RNA Polymerase (RdRp) that replicates and transcribes the RNA genome through a double stranded RNA (dsRNA) intermediate [24]. As part of the evolutionary arms race between viruses and host, mammalian cells are equipped with cytosolic sensors for the detection and degradation of large viral dsRNA [25]. To avoid detection, it is thought that members of this virus group have evolved to distort cellular membranes for the construction of immune system-naive membrane invaginations, called vesicle packets, that facilitate viral replication despite surveillance by the innate immune sensors [26]. Within these viral compartments, viral RNA is synthesized and exported through the lumen of the vesicle packet and into the cytoplasm for the translation of viral proteins [27]. Although the precise molecular details underlying vesicle packet formation are not fully understood, the recruitment of ESCRTs to sites of replication has been established for several emerging mosquito-borne flaviviruses including Dengue virus, all of which are thought to be facilitated through the viral helicase protein NS3 [28,29]. However, only in the case of Yellow Fever Virus replication was a canonical L-Domain sequence identified. In this scenario, the upstream ESCRTs may act in a membrane scaffolding capacity where they serve to keep the vesicle packet lumen open to facilitate the export of viral RNA transcripts and genomes into the cytoplasm. Lack of obvious L-Domain sequences despite clear biochemical association raises intriguing questions, about how dynamic ALIX recruitment is to sites of viral assembly on membranes, and whether or not other enveloped viruses of clinical importance, like Herpes Simplex Virus 1 (HSV-1), harbor biologically relevant ALIX recruitment L-domains within their structural proteins to facilitate essential membrane-associated steps in their life cycles [30].

Unlike RNA viruses, particle formation of the DNA virus HSV-1 begins in the nucleus (Figure 2b) [31]. Therein, after having undergone lytic reactivation, the newly synthesized viral DNA is packaged within the viral capsid [32]. In order to produce infectious progeny virions, it is theorized that the nuclear viral capsid must pass through the nuclear envelope through a membrane budding pathway, since HSV-1 capsids are too large to traverse the nuclear pore complex [33]. To do this, herpesviruses assemble a nuclear egress complex (NEC) from viral UL31 and UL34 proteins (HSV-1 nomenclature) [30]. UL34 localizes to the inner nuclear envelope via its C-terminal type II single-pass membrane anchor [34]. UL31 binds to HSV-1 capsid vertices but also associates with membrane bound UL34 and effectively acts as a dynamic scaffold between the inner nuclear membrane and the viral capsid [31]. Additionally, the UL31-UL34 heterodimer forms a lattice about the inner nuclear membrane in a manner structurally reminiscent of HIV-1 Gag assembly sites (Figure 2a,b) [31,35,36]. In order to efficiently bud across the nuclear envelope, HSV-1 and Epstein-Barr Virus (EBV) were recently shown to recruit the early acting ESCRT protein ALIX and ESCRT-III associated protein VPS4 directly to the NEC via interactions with UL34 [37,38]. Remarkably, EBV BFRF1 (UL34 in HSV-1), was also shown to recruit the NEDD4-like ubiquitin ligase ITCH, and that ubiquitination of BFRF1 regulates the formation of nuclear envelope-derived vesicles during virus maturation, drawing yet another parallel with retroviral budding [39]. Depletion of ALIX resulted in the loss of the downstream ESCRT-III component required for membrane scission, CHMP4B, and reduced HSV-1 and EBV replication. Additionally, in the case of HSV-1, impairment of the membrane scission machinery VPS4 resulted in a significant accumulation of enveloped virions in the nucleus, some of which remained tethered to the inner nuclear membrane [37], suggesting a role for ESCRTs in herpesvirus nuclear egress. However, despite clear biochemical data describing physical interactions between ALIX and UL34, no L-Domain sequence was identified. The remainder of this review focuses on comparing the nuclear egress of HSV-1 capsids to that of retroviral budding sites and proposes the existence of an ALIX binding site within UL34 that could support the model of ALIX recruitment to herpes-viral NECs.

## 2. A Hexagonal Assembly of HIV-1 Gag Drives Particle Maturation and ESCRT Recruitment

HIV-1 particle assembly at the membrane has been the focus of intensive study for more than two decades [5]. The retroviral Gag protein traffics to the membrane and forms higher-order oligomers via its capsid domain and assembles into an immature lattice that exhibits a hexagonal geometry (Figure 3a) [40]. The Gag lattice is intrinsically curved which is thought to assist in the shaping of the plasma membrane into a spherical membrane bud [41]. Maturation of the lattice is an essential step in the production of infectious virus particles and is still to this day not fully understood [42]. What is known is that the viral genome along with the viral proteins required for maturation are packaged as a ribonucleoprotein (RNP) within the center of the immature lattice, and packaging of the RNP is what triggers Gag maturation [18]. During or after this step, the upstream ESCRT proteins are recruited to the lumen of the retroviral budding neck via interactions with the Gag p6 domains located at the cytoplasmic face and luminal opening to the immature lattice [43]. ESCRT-I then assembles into a ring that templates the oligomerization of downstream ESCRT-III and the membrane scission machinery which abscises membranes by forming filaments within membrane necks [18,44,45,46]. ALIX is also an essential component of this process and exhibits the same localization pattern, as does ESCRT-I [47]. By binding to a spatially separated L-domain from that of ESCRT-I, ALIX can localize to the budding neck lumen and recruit ESCRT-III directly by virtue of its p6-binding V domain and CHMP4B-binding Bro1 domain, respectively [47,48,49]. Evidence of these structures have been captured in relevant cell types as well as in vitro [44,50,51]. Although separate in vitro studies have revealed that Gag assembly alone may be sufficient to drive membrane curvature of reconstituted vesicles, [52] the presence of L domain sequences is a requirement for proper retroviral assembly and release in cells [5].

Retroviral budding is a highly organized process in which the formation of the Gag lattice is critical for the proper recruitment and function of ESCRT in virus release [44,53]. Although ESCRT-I is transiently recruited with Gag to sites of active assembly, its functional significance only comes into play as an aggregate of individual recruitment events [46,50]. Indeed, it is only after recruitment of TSG101 or ALIX that ESCRT-III can be recruited and induce membrane scission [54]. This was shown recently through disruption of the ESCRT-I helical interface that left autophagosomes unsealed and prevented the release of HIV-1, indicative of a lack of downstream recruitment and scission [44]. This suggested that the budding site, and more importantly the hexagonal lattice, of HIV-1 is a well-evolved molecular signature that topologically mimics the cellular recruitment signal behind ESCRT function in cellular cargo trafficking pathways [55].

The immature hexagonal lattice exhibited by HIV-1 Gag during assembly is such an essential component of the particle maturation pathway that it has even become the target of novel antiretroviral drugs [56]. Bevirimat, the first maturation inhibitor, was designed to block the complete maturation of HIV-1 Gag such that the Gag-derived capsid hexamers that ultimately assemble into the core cannot be liberated from the Gag polyprotein [56]. Assembly sites targeted by this drug exhibited thick and mostly unprocessed Gag shells that remained associated with the membrane and packaged the genome eccentrically [56]. Additional inhibitors have been raised against the luminal opening of the Gag lattice [57]. More specifically, it was shown that HIV-1 assembly could be inhibited by targeting the N-terminal domain of TSG101 with the drug N16 that binds p6 domains within the Gag hexagonal lattice [57]. Intriguingly, in determining the mechanism of action of N16, it was found that this compound disrupted interactions outside of the p6 binding site in TSG101 [57]. The N-terminus of TSG101 contains an ancestral and inactivated domain that may have once been used as part of the ubiquitin conjugation pathway [58]. Despite having lost its catalytic ability, the TSG101 N-terminus still retains the ability to bind ubiquitin. N16 impairs HIV-1 maturation by binding to this secondary ubiquitin binding site, as opposed to the p6 binding pocket [57]. Given that ubiquitin conjugation to Gag is essential for ESCRT-I mediated virus release [17], the dramatic effect of N16 on virus release underscores the importance of ubiquitin in the process of virus particle maturation.

## 3. The NECs of Herpesviruses form Gag-Like Assemblies That Interact with ALIX and Are Required for Efficient Nuclear Egress

Unlike in retroviral budding where infectious particles are released from the plasma membrane, HSV-1 particle assembly and egress occurs first at the inner nuclear envelope and then again at the plasma membrane, for the release of infectious particles [32]. This first process is carried out by two viral proteins UL31 and UL34 which form the inner nuclear membrane-associated NEC [59]. UL34 anchors this complex to the membrane via a C-terminal transmembrane domain where it recruits the capsid-binding UL31 protein and forms a stable heterodimer [59]. Recently, it was determined through high resolution structural studies that multiple copies of the UL31-UL34 heterodimer assemble into a retroviral Gag-like hexagonal lattice on the surface of the inner nuclear membrane (Figure 3b) [35,36,60]. Interestingly, like HIV-1 Gag, in vitro studies using purified NECs on reconstituted membranes showed that the NECs themselves are sufficient to drive membrane deformation and egress [60]. The relevance of these higher order structures in virus replication was validated via mutagenesis in which inter-hexamer contact sites were abrogated [61]. In doing so, it was proposed that the hexagonal lattice plays an important role in regulating primary envelopment, and therefore viral replication [31,61]. Additionally, there are notable cellular binding partners that also promote herpesvirus primary envelopment including the WD repeat-containing protein 5 (WDR5) and the ESCRT-associated protein ALIX [37,62]. ALIX was shown to interact with both recombinant UL34 and the UL34 from HSV-1-infected cell lysate, an interaction that drives nuclear virus egress in cells [37]. In parallel, disruption of the membrane scission machinery VPS4 by mutagenesis also reduced virus nuclear egress [37]. In contrast, one previous study showed that in a different cell line, HSV-1 nuclear egress was unaffected by VPS4 inactivation [63]. This hints at a potential cell-type specific involvement of ESCRT proteins in virus nuclear egress. Intriguingly, both Bro1 and V domains of ALIX were capable of interacting with UL34, hinting at the possible presence of an L-domain sequence, but also that there may be additional electrostatic interactions between the proteins as is seen in ALIX interactions with HIV-1 Gag [64,65]. However, although HSV-1 harbors almost two dozen potential ESCRT-I TSG101 and ALIX binding sites across its viral proteins [66], TSG101 and ALIX do not appear to be essential for infectious virus release despite both being detected within extracellular virions [66]. Further still, no ALIX binding sites were identified in either component of the NEC which complicates the interpretation of ALIX–NEC interactions [66,67]. One possible explanation for the results of these studies are that their sequencing surveys relied on canonical L-domain sequences that are typically attributed to ESCRT and ALIX interactions based on the most well characterized cellular and viral binding partners [15,21,68]. Regardless, more evidence is needed before any definitive role for ESCRTs in nuclear egress can be established. 

In analyzing the NEC for potential non-canonical L-domains, like those found in the RNA viruses mentioned above that may promote its association with ALIX, sequence alignments and structural modelling were used to uncover such potential motifs. The sequence of HSV-1 UL34 was compared to that of several herpesviruses across three herpesvirus families (alpha, beta, and gamma-herpesviruses) (Figure 4a). Sequence alignments revealed little homology across virus families and family members. HSV-1 and HSV-1-2, both alpha-herpesviruses, share almost 80% identity, while the same protein from Varicella zoster virus (VZV), another alpha-herpesvirus, had less than 40% sequence identity with HSV-1. Additionally, less than 15% identity was seen between HSV-1 and members of the beta-herpesviruses family like Human Cytomegalovirus (HCMV), Human Herpesvirus 7 (HHV7), and Murine Cytomegalovirus (MCMV), as well as the gamma-herpesvirus family members Kaposi’s Sarcoma-Associated Herpesvirus (KSHV) and EBV. Three putative L-domain motifs were identified in HSV-1 and were highlighted in yellow (Figure 4a). The sequences were also placed in a table along with known viral L-domain motifs (Figure 4b). The first potential L-Domain sequence was the near N-terminal sequence _8_YPGHPGDAFEGL_19_. However, despite the predicted flexibility of this sequence, it sterically clashed with UL31, suggesting that it would not likely be accessible for binding to ALIX, or could be engaged in interactions with UL31. The second sequence just downstream was _41_YSPSSL_46_. However, this motif was shown to embed its tyrosine residue within secondary structure elements of UL34 where it makes hydrophobic contacts with neighboring phenylalanine and tryptophan residues in the context of the UL31-UL34 heterodimer, making it inaccessible for binding to ALIX [31]. The third and most likely candidate for an ALIX-binding L-domain sequence was _202_YGAEAGL_208_. However, this region of UL34 has not been previously characterized structurally, likely due to its intrinsic flexibility and location proximal to the C-terminal transmembrane domain. Therefore, a molecular model of this region of the HSV-1 NEC was generated using the resolved residues from the HSV-1 NEC crystal structure (PDB:4ZXS), combined with structural predictions from Phyre2 and MODELLER in order to structurally analyze the missing regions of the protein [31,69,70] (Figure 4c). The prediction of this flexible region revealed the presence of a loop that faces away from the rest of the NEC and contains the putative L-Domain sequence _202_YGAEAGL_208_. Curiously, sequence alignments revealed little conservation of putative L-Domains across viruses, suggesting that the model for ALIX recruitment may only occur in select herpesvirus nuclear egress pathways. Perhaps in cases of ESCRT-III dependent nuclear egress, an edge of the NEC containing the L-Domain sequence is exposed to the nucleoplasm to be recognized for ALIX-mediated membrane budding and scission, much like how HIV-1 Gag recruits ALIX to the plasma membrane [71]. Although this model places the L-Domain of the NEC proximal to the nuclear envelope, this proposed mechanism is not unlike the recruitment of ESCRTs seen for beta-retroviruses and spuma-retroviruses where ESCRT binding is predicted to occur close to the plasma membrane [72,73]. This model could serve as a guide for future studies looking to further our understanding of the molecular basis and relevance behind ALIX recruitment and function at sites of viral nuclear egress.

## 4. ESCRT Is Recruited to Sites of RNA and DNA-Protein Complex Egress via a Shared Molecular Signature

The parallels between retroviral budding and viral nuclear egress described here point toward a remarkable replication mechanism reached by divergent virus families. These viruses operate via a particle assembly pathway that is topologically equivalent to cellular mechanisms [74,75,76]. Using the assembly components of either of these viruses as tools to study the role of the ESCRTs in these processes could help address longstanding questions in the field of membrane biology as to why some damaged membranes are reparable and sealable, while others are recycled, both via an ESCRT-dependent mechanism [76,77]. Simultaneously, these studies may also uncover previously overlooked mechanisms of how viruses selectively recruit the ESCRT machinery at very discrete steps to avoid being recycled themselves [12]. More specifically, viral mechanisms of antagonism of regulatory proteins associated with the ESCRTs and their intersection in cellular pathways such as autophagy could be identified [78,79].

One example of a scenario in which the nuclear envelope is recycled rather than repaired is in the membrane damage induced by defects in Lamin A in cases of Progeria [76]. Progeria is a disease caused by the accumulation of a truncated form of Lamin A that severely compromises nuclear architecture [80]. In the analysis of patient samples with clinical manifestations of the disease, it was shown that ALIX was the principal recruitment factor for the autophagy machinery used for the degradation of membrane-damaged areas, rather than for the repair of these areas. This study did not investigate the molecular basis behind ALIX recruitment to these sites, but proposed a mechanism by which excess membrane protein is sensed and marked for degradation via ALIX as part of nuclear envelope quality control surveillance [76]. Lamins readily adopt intermediate filaments as part of the nuclear cytoskeleton [81]. The truncation seen in Progeria occurs in the C-terminal portion of the protein which does not compromise the ability of the pathological Lamin A variant, progerin, from forming filaments, but may alter its ability to associate with DNA [82]. Curiously, lattice-like intranuclear aggregates were observed when Lamin A or a Lamin A D446V mutant from Polish patients with Emery-Dreifuss muscular dystrophy type 2 (EDMD2) were over-expressed [83]. Nuclear blebbing is also a frequently observed phenotype due to progerin expression or when other Lamin proteins are depleted [84]. Further, these nuclear blebs appear to be enriched in euchromatin, despite being transcriptionally inactive [85]. The phenotypes observed in these conditions raises the question of whether or not aberrantly associated genomic nucleic acid is also a component of these defective Lamin A assemblies that are sensed and degraded via the ESCRT pathway. Evidence for this association can be seen in the genomic instability and impaired DNA damage response seen in Progeria patients [86,87]. This quality control process could be another instance in which a nucleic acid-associated protein assembly distorts a membrane compartment which triggers ALIX or ESCRT-III recruitment.

In a phenotypically normal scenario, large RNPs required for neuronal plasticity can be seen budding across the inner nuclear membrane of *Drosophila* cells in a similar manner to HSV-1 capsids [88]. Like HSV-1 particles, these RNA-protein aggregates are too large to passively diffuse through the nuclear pore complex and require the assistance of the ESCRT proteins to bud through the inner nuclear membrane [37]. However, only the downstream ESCRT-III protein Shrub (CHMP4B in humans) has been found to regulate this process [37,89]. Further, the current understanding of these RNPs points to their assembly as granules as opposed to highly organized lattices like that seen by HSV-1 NEC or HIV-1 Gag [90]. Yet these structures represent another example of a membrane bound RNP that triggers ESCRT recruitment. Since ALIX has been shown to recruit Shrub during cell division in *Drosophila* germline cells, it plausible that ALIX could also serve as a major recruitment factor of Shrub for large RNP nuclear egress [91] as well. Like with HSV-1 nuclear egress, more insight is needed before the role of ESCRTs in large RNP nuclear egress is determined with certainty.

## 5. Conclusions

Despite the tremendous diversity seen across their genomes and morphologies, viruses are bound by the molecular machinery of their cellular hosts. The outcome is the evolution of several independent viral mechanisms that converge on cooption of conserved host mechanisms [92]. One of the most well studied of these pathways hijacked by viruses is the endosomal sorting pathway which is orchestrated by the ESCRT proteins [8,75]. While HIV-1 requires this cellular machinery for the last step in its replication cycle, is proposed to hijack these proteins at a much earlier step in the viral replication cycle stage that is distinct from HIV-1 [37,66]. In parallel, flaviviruses coopt the ESCRTs in yet a third way to sites of replication in the ER [12]. In all three cases, it appears that the role the ESCRTs play is to facilitate the formation and subsequent abscission of a budding neck structure that is essential to the replication of the virus utilizing it [16]. However, the nature by which these assembly sites are established and maintained remains obscure. What has become clear from recent structural studies of HIV-1 and HSV-1 is that the hexagonal lattices formed during particle maturation for both of these viruses bear striking structural and mechanistic similarities [31,40,52,60]. Whether they be in the nucleus as part of the ESCRT-mediated model of nuclear egress for HSV-1 or at the cytoplasmic periphery during the final moments of retroviral budding, these lattices represent fundamental convergence on a mechanism used by viruses for usurpation of a cellular pathway [16,61,77]. These structures operate with no consensus sequence or structure but more likely create a precise molecular signature within membrane compartments that is ideal for ESCRT recruitment. Numerous insights pertinent to health and disease processes are to be gained from studying assemblies such as these so that we may better our understanding of the fundamental cellular pathway of membrane trafficking. Indeed, work is already being done aimed at understanding the damage caused by the aggregation of proteins such as tau in cases of Alzheimer’s disease [93]. In an attempt to mitigate the cytotoxicity of the prion-like protein, tau, the ESCRTs, and ALIX, are involved in the recycling of tau via autophagy [93]. However, tau aggregates are capable of breaching this membrane barrier in a manner irreparable by the ESCRT machinery, thereby increasing tau aggregation propensity and worsening disease states [93]. The molecular insights gained from studying the sealing of virus particles could help answer fundamental questions about membrane biology and improve our understanding of why the damage inflicted by tau on the endosome is un-sealable. Furthering our understanding of these mechanisms can potentially lead to the generation of novel therapeutics that promote tau clearance and mitigate disease, or interfere with virus assembly pathways.

## Figures and Tables

**Figure 1 viruses-13-01138-f001:**
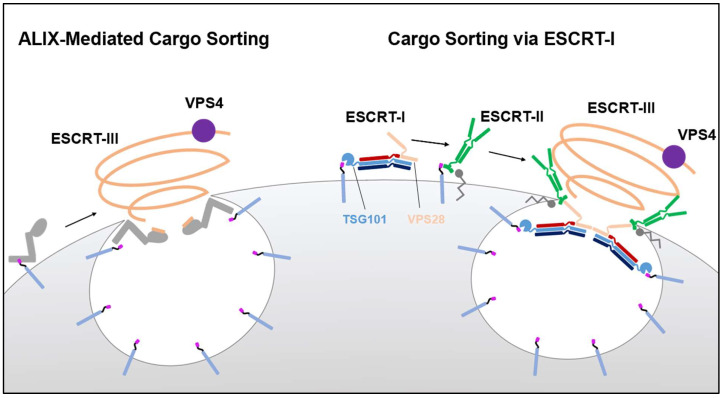
The respective roles of ALIX and ESCRT-I in the sorting of membranous cargo. Upon internalization, ubiquitylated cargo is detected by ALIX (**left**) and ESCRT-I (**right**) for compartmentalization into intraluminal vesicles that are destined for degradation via the late endosome. Both ALIX and ESCRT-I contain ubiquitin binding domains that facilitate this first step. Unlike ESCRT-I, ALIX possesses an ESCRT-III binding domain that allows for the direct recruitment of ESCRT-III and VPS4, the machinery required for sealing of cargo within intraluminal vesicles and abscising these vesicles from the endosomal membrane. In a similar fashion, the ESCRT-I component TSG101 binds ubiquitylated cargo, while the VPS28 component can recruit ESCRT-III through ESCRT-II which also binds ubiquitylated cargo as well as phospho-inositol lipids.

**Figure 2 viruses-13-01138-f002:**
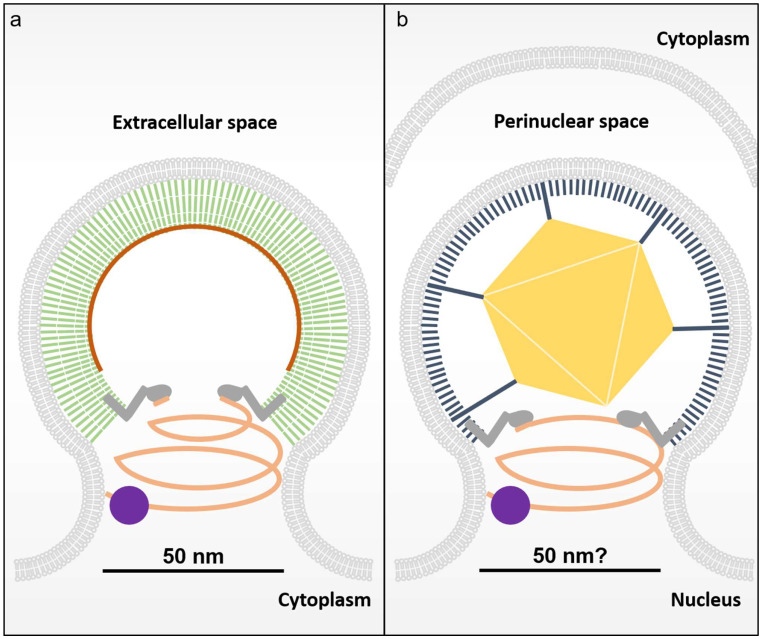
The process of HSV-1 nuclear egress and scission resembles the retroviral egress pathway. (**a**) Diagram of the well characterized HIV-1 budding site. HIV-1 Gag (green sticks) binds RNA (dark orange) and assembles on the membrane and recruits ALIX (dark grey) via the L-Domain located in its C-terminal p6 domain. HIV-1 budding necks have been shown typically to adopt a diameter of approximately 50 nm prior to membrane scission and this topology is likely to be shared by the NEC given the mechanistic constraints of the ESCRT-III membrane scission machinery. The downstream ESCRT-III is shown in salmon and the VPS4 membrane scission machinery in purple. (**b**) HSV-1 particle assembly begins in the nucleus where a newly formed capsid must transit the nuclear envelope by assembly and budding. Schematic of the HSV-1 nuclear egress complex prior to membrane scission assembled with its capsid as it buds through the nuclear envelope. A single virus capsid is shown as a gold hexagon and the NEC UL31-UL34 heterodimer is shown as membrane-associated sticks (black) that occasionally interact with the vertices of the capsid. ALIX is recruited to sites of nuclear egress through interactions with NEC and mediates the recruitment of the downstream ESCRT components to facilitate membrane scission.

**Figure 3 viruses-13-01138-f003:**
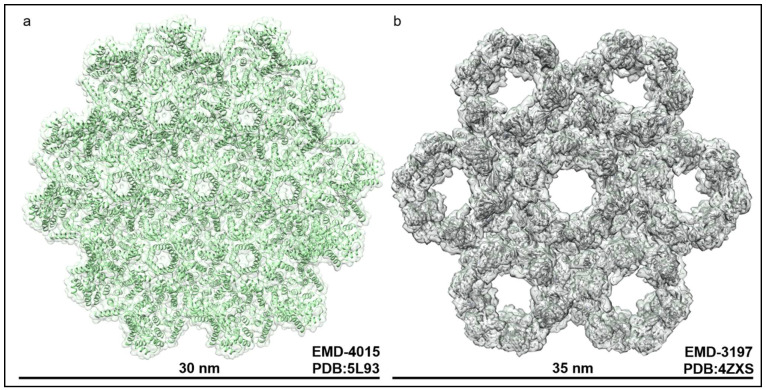
Both HSV-1 and HIV-1 form membrane bound hexagonal lattices that are involved in envelopment. HIV-1 Gag forms higher-order assemblies on the plasma membrane during viral budding. (**a**) The immature Gag lattice in green adopts a hexagonal geometry when bound to the plasma membrane and is found within immature virus particles. This viral complex must assemble on membranes in order to trigger the recruitment and activity of the membrane scission and sealing machinery. (**b**) The HSV-1 NEC complex binds to the inner nuclear membrane and assembles into a hexagonal lattice containing repeating units of a UL31-UL34 heterodimer. Structure of the HSV-1 NEC by fitting of the NEC crystal structure (PDB code: 4ZXS) into the 3D average of the NEC coat that was recently resolved via cryo-electron microscopy for Pseudorabies virus [36]. One repeating unit of the hexagonal lattice is shown, which has a diameter of approximately 35 nm. Note the differences in inter-subunit compaction and diameters between the assemblies.

**Figure 4 viruses-13-01138-f004:**
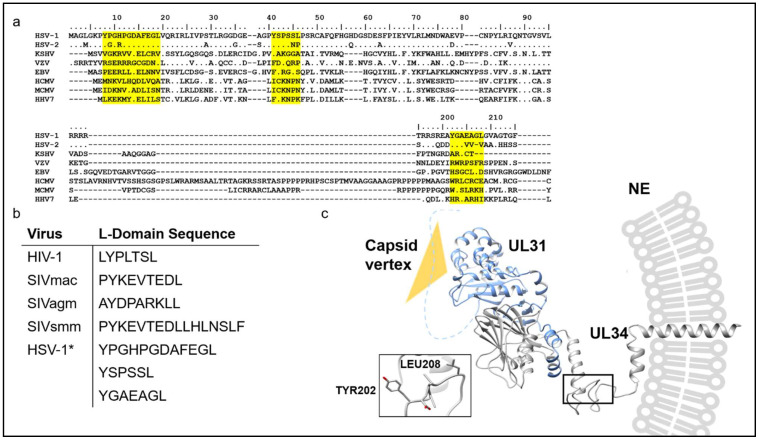
Sequence and structure analysis of HSV-1 UL34 reveals a putative L-Domain-like sequence that may promote its interactions with ALIX. (**a**) Sequence alignment of the HSV-1 NEC complex component UL34 across herpesvirus families. Putative L-Domain residues seen in HSV-1 are highlighted in yellow. (**b**) Sequence comparison between previously identified L-Domains with those identified from the sequence analysis herein (marked with an asterisk). (**c**) Molecular modeling of the membrane-bound NEC complex from HSV-1. The presence of a potential ALIX binding site located on a flexible inter-domain linker is shown boxed and with an inset. This linker region within UL34 contains a tyrosine residue (TYR202) that extends outward and away from the rest of the protein. At the edges of the NEC hexagonal assembly, these flexible loops could be exposed to mediate recruitment of ALIX to sites of nuclear egress.

## Data Availability

All molecular models and data generated during the construction of this review are available upon request.

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
