# Peer review of "When in Need of an ESCRT: The Nature of Virus Assembly Sites Suggests Mechanistic Parallels between Nuclear Virus Egress and Retroviral Budding"

_viruses, 2021, doi:10.3390/v13061138_

Round 1

Reviewer 1 Report

  1. The author should present their model of nuclear egress throughout the manuscript as a hypothesis rather than a certainty. Whereas the mechanism by which ESCRT-III and other ESCRT components mediate scission during HIV budding are well studied, the same cannot be said about HSV nuclear egress.

  1. The author has expanded the explanation of ESCRTs, but readers who are not experts would benefit from a diagram that shows different components of the machinery (ESCRTs I, II III, ALIX, and TSG101) and illustrates their respective roles in cellular budding and scission.

  1. Lines 15-16: “For instance, at the plasma membrane, ESCRT recruitment is essential for HIV-1 assembly and budding, while it is dispensable for the release of HSV-1.” In fact, ESCRTs are required for cytoplasmic budding of HSV-1 (see Crump et al JVI 2007 and Pawliczek et al JVI 2009).

  1. One of the key elements of the author’s model is that the L domain in UL34 recruits ALIX. The L domain in HIV Gag is located at its membrane-distal end, which would make ALIX recruitment straightforward. By contrast, the proposed L domain in UL34 is located within the membrane-proximal region. How does the author envision ALIX recruitment given the likely geometric constraints?

  1. The sequence alignment in Figure 3 will be easier to understand with larger font and fewer colors.

Author Response

  1. The author should present their model of nuclear egress throughout the manuscript as a hypothesis rather than a certainty. Whereas the mechanism by which ESCRT-III and other ESCRT components mediate scission during HIV budding are well studied, the same cannot be said about HSV nuclear egress.

The author greatly appreciates the Reviewer’s comments on how to improve the reader’s interpretation of biological data in the text. On line 248 “revealed” was replaced with “proposed” when discussing biochemical data about the role of the NEC in HSV-1 replication. Additionally, “it is theorized that” was inserted into line 130 after describing the process of nuclear egress for the first time. The title of Figure 2 was changed to “The proposed process of HSV-1 nuclear egress and scission resembles the retroviral egress pathway” on line 156. These three alterations were made to the text as an effort to remind the reader that the involvement of ESCRTs in HSV-1 nuclear egress is far from certain.

2. The author has expanded the explanation of ESCRTs, but readers who are not experts would benefit from a diagram that shows different components of the machinery (ESCRTs I, II III, ALIX, and TSG101) and illustrates their respective roles in cellular budding and scission.

The author agrees with the comments from the Reviewer regarding the introductory section. In order to aid non-specialists in understanding the relative roles of the ESCRT proteins, an additional figure (now figure 1) was inserted into the text. Now on line 118, the following figure legend was inserted that compares the respective roles of ALIX and ESCRT-I in cargo trafficking and scission: “Figure 1: The respective roles of ALIX and ESCRT-I in the sorting of membranous cargo. Upon internalization, ubiquitinated cargo is detected by ALIX (left) and ESCRT-I (right) for compartmentalization into intraluminal vesicles that are destined for degradation via the late endosome. Both ALIX and ESCRT-I contain ubiquitin binding domains that facilitate this first step. Unlike ESCRT-I, ALIX possesses an ESCRT-III binding domain that allows for the direct recruitment of ESCRT-III and VPS4, the machinery required for sealing of cargo within intraluminal vesicles and abscising these vesicles from the endosomal membrane. In a similar fashion, the ESCRT-I component TSG101 binds ubiquitinated cargo, while the VPS28 component can recruit ESCRT-III through ESCRT-II which also binds ubiquitinated cargo as well as phosphoinositol lipids.”

3. Lines 15-16: “For instance, at the plasma membrane, ESCRT recruitment is essential for HIV-1 assembly and budding, while it is dispensable for the release of HSV-1.” In fact, ESCRTs are required for cytoplasmic budding of HSV-1 (see Crump et al JVI 2007 and Pawliczek et al JVI 2009).

 The author appreciates the Reviewer’s critical review of the abstract. In order to clarify the relative ESCRT recruitment pathways and how they differ between virus species, the following changes were made to the abstract. First, on line 13, “ESCRT” was replaced with “ESCRT-I, -II, and -III” to introduce the idea that separate ESCRT complexes exist. Second, on line 15, “ESCRT” was replaced with “ESCRT-I” and “ESCRT” on line 17 was replaced with “ESCRT-III.” These changes clarify that ESCRT-I is dispensable for HSV-1 particle egress from the plasma membrane while it is required for the release of HIV-1. Additionally, these changes clarify that ESCRT-III precisely is required for HSV-1 nuclear assembly and egress. The references mentioned by the Reviewer have already been incorporated into the text and discussed.

4. One of the key elements of the author’s model is that the L domain in UL34 recruits ALIX. The L domain in HIV Gag is located at its membrane-distal end, which would make ALIX recruitment straightforward. By contrast, the proposed L domain in UL34 is located within the membrane-proximal region. How does the author envision ALIX recruitment given the likely geometric constraints?

 The author appreciates this excellent question posited by the Reviewer. To address this, the following text was inserted on line 305: “Although this model places the L-Domain of the NEC proximal to the nuclear envelope, this proposed mechanism is not unlike the recruitment of ESCRTs seen for betaretroviruses and spumaretroviruses where ESCRT binding is predicted to occur close to the plasma membrane.” This insertion also includes the following references:

Gottwein, Eva et al. “The Mason-Pfizer monkey virus PPPY and PSAP motifs both contribute to virus release.” Journal of virology vol. 77,17 (2003): 9474-85. doi:10.1128/jvi.77.17.9474-9485.2003

Stange, Annett et al. “Characterization of prototype foamy virus gag late assembly domain motifs and their role in particle egress and infectivity.” Journal of virology vol. 79,9 (2005): 5466-76. doi:10.1128/JVI.79.9.5466-5476.2005

These additional references argue in favor of ESCRT recruitment to membrane proximal sites given that the location of the ESCRT recruitment domains within the Gag proteins for each of these retroviral species (betaretroviruses and spumaretroviruses) is substantially closer to the membrane compared to HIV-1.

5. The sequence alignment in Figure 3 will be easier to understand with larger font and fewer colors.

The author agrees with the Reviewer regarding the suggested improvements for Figure 3 (now Figure 4). The alignment shown has been replaced by a version with slightly larger font and is now uncolored.

Reviewer 2 Report

In this revised manuscript, the author has addressed queries raised in previous review. I have only minor comments that are listed below:

1) Line 8 abstract. Viruses in my view do not have goals as that terms suggests some cognitive effort. I would use a better word like process

2) Line 30 introduction. I have not heard the term "encapsulated" to described membrane acquisition of endosomal cargo. Maybe it is in common usage but I don't think it is a standard term.

3) Line 325. The sentence states "was so far found..."  "was so far found" is an odd construction

Author Response

1) Line 8 abstract. Viruses in my view do not have goals as that terms suggests some cognitive effort. I would use a better word like process

The author welcomes the Reviewer’s suggestion for improving the clarity of the abstract. The word “goal” on line 8 has been replaced with “step.”

2) Line 30 introduction. I have not heard the term "encapsulated" to described membrane acquisition of endosomal cargo. Maybe it is in common usage but I don't think it is a standard term.

The author thanks the Reviewer for their feedback in this section of the introduction. The word “encapsulated” on line 30 has been replaced with “compartmentalized.”

3) Line 325. The sentence states "was so far found..."  "was so far found" is an odd construction

The author appreciates the Reviewer’s suggestion to improve the clarity of the statement in the text. The referenced phrase on line 325 has been replaced with “has been found.”

This manuscript is a resubmission of an earlier submission. The following is a list of the peer review reports and author responses from that submission.

Round 1

Reviewer 1 Report

 Manuscript Review:

 Manuscript: In need of an ESCRT: The structures of virus assembly sites reveal mechanistic parallels between nuclear egress and retroviral budding

 Author: Rose, KM

 Journal:    Viruses

 The author has provided a brief review of RNA virus particle budding and nuclear egress of the alpha-herpesvirus, HSV as a forum to draw parallels between these processes. The conclusion from this review is that there is a common cellular process that could be responsible for these two essential steps in virus assembly that are spatially divergent.

 Overall the review draws from more recent literature and frames the shared mechanisms of membrane deformation that accompany both processes. Of interest, it raises the concept of membrane damage and repair as a potential central tenet in both RNA virus budding and nuclear egress of HSV. 

 Although the review is informative, it will be difficult for many readers of Viruses to relate the vast amount of data that surrounds the description of this critical step in virus assembly to the authors’ arguments. This perhaps can be explained by the lack of well laid foundation for the processes of RNA virus budding and, HSV nuclear egress. Both of these sections were given less emphasis because of the authors’ interest in a discussion of late domains and predicted late domains in HSV.  In addition, some of the wording is confusing. As an example in the abstract the statement “For instance, at the plasma membrane, ESCRT recruitment is essential for HIV assembly and budding, while it is indispensable for the release of HSV”.  A universally accepted role of the ESCRT in HSV envelopment and release has not been reported and remains controversial, thus this statement in the abstract suggests that it is also essential for HSV. More recent data has argued for components of the ESCRT-III complex in cytoplasmic envelopment of HSV but even this data has been challenged by some investigators.  Furthermore, the first statement in section 3 indicates that HIV infected cells release infectious particles from the plasma membrane whereas “HSV particle assembly and egress occur at inner nuclear envelope”. My understanding is that the nuclear envelope contains an inner and outer membrane that HSV buds through these membranes during its final envelopment in the cytoplasm. Although the author may be describing only the process at the inner nuclear membrane, the statement in the text is far from clear could be interpreted that infectious HSV buds from the nuclear envelope. 

 Throughout the manuscript the text contained statements of fact with no references, including in the figure legends (see for example fig 3 legend). This was distracting when trying to link statements with specific publications. Lastly, the manuscript requires significant editing. As examples in the Introduction, the term African green (I assume monkey); last part of section 2, “This drug impairs HIV maturation (I assume drug mentioned at the top of the paragraph but not sure).

 In summary, I found this review interesting and provocative but I believe that the current version is not suitable for publication. At a minimum the review should provide  a clearer description of the fundamental processes of HIV budding at the plasma membrane and HSV nuclear egress in order to frame the concept the author is trying to convey. 

Author Response

Although the review is informative, it will be difficult for many readers of Viruses to relate the vast amount of data that surrounds the description of this critical step in virus assembly to the authors’ arguments. This perhaps can be explained by the lack of well laid foundation for the processes of RNA virus budding and, HSV nuclear egress. Both of these sections were given less emphasis because of the authors’ interest in a discussion of late domains and predicted late domains in HSV.  In addition, some of the wording is confusing. As an example in the abstract the statement “For instance, at the plasma membrane, ESCRT recruitment is essential for HIV assembly and budding, while it is indispensable for the release of HSV”.  A universally accepted role of the ESCRT in HSV envelopment and release has not been reported and remains controversial, thus this statement in the abstract suggests that it is also essential for HSV. More recent data has argued for components of the ESCRT-III complex in cytoplasmic envelopment of HSV but even this data has been challenged by some investigators.  Furthermore, the first statement in section 3 indicates that HIV infected cells release infectious particles from the plasma membrane whereas “HSV particle assembly and egress occur at inner nuclear envelope”. My understanding is that the nuclear envelope contains an inner and outer membrane that HSV buds through these membranes during its final envelopment in the cytoplasm. Although the author may be describing only the process at the inner nuclear membrane, the statement in the text is far from clear could be interpreted that infectious HSV buds from the nuclear envelope. 

I thank the reviewer for their helpful comments and suggestions for improvement of the manuscript. Several areas of the manuscript have been extended to address the reviewer’s concerns. First, line 16 in the abstract has been modified to correct the statement: “ESCRT recruitment is essential for HIV assembly and budding, while it is dispensable for the release of HSV.” Secondly, the statement at the beginning of section 3 has been modified on line 184: “Unlike in retroviral budding where infectious particles are released from the plasma membrane, HSV particle assembly and egress occurs first at the inner nuclear envelope and then again at the plasma membrane, for the release infectious particles.”

Throughout the manuscript the text contained statements of fact with no references, including in the figure legends (see for example fig 3 legend). This was distracting when trying to link statements with specific publications. Lastly, the manuscript requires significant editing. As examples in the Introduction, the term African green (I assume monkey); last part of section 2, “This drug impairs HIV maturation (I assume drug mentioned at the top of the paragraph but not sure).

Several references were added throughout to support claims made in the text. For instance, in lines 30-34, an introduction to ESCRT function was added with several references to support these claims. Additionally, section 2 line 156 addresses the identity of the drug in reference with additional mentions on lines 157 and 161 to guide the reader more clearly. ‘Monkey’ was added to line 48 to clarify the organism being mentioned.

 In summary, I found this review interesting and provocative but I believe that the current version is not suitable for publication. At a minimum the review should provide  a clearer description of the fundamental processes of HIV budding at the plasma membrane and HSV nuclear egress in order to frame the concept the author is trying to convey. 

The reviewer’s feedback was very helpful and informative in improving the manuscript. Lines 30-33, 62-64, and 69-72 were added to the manuscript to provide sufficient biological descriptions to the viral mechanisms of replication being mentioned. Additionally, lines 186, 195, 200, and 206 were amended to add more detail to assist the reader in understanding the role of the HSV NEC in the virus lifecycle.

Reviewer 2 Report

Summary

Viruses co-opt cellular machinery to aid in their replication at different stages in their replication cycles. Both HIV and HSV-1 have been shown to recruit Endosomal Sorting Complex Required for Transport (ESCRT) proteins to facilitate membrane scission during viral egress from infected cells. While HIV requires ESCRT recruitment to pinch off from the plasma membrane as a final exit step, HSV-1 utilizes ESCRT proteins to pinch off from the inner nuclear membrane during the early stages of egress. This review attempts to draw mechanistic parallels between the two processes, which is both interesting and timely. However, the overstatement of the mechanistic similarities, the somewhat cursory introduction of the ESCRTs, and the lack of clarity in the text and figures detract from the central message.

Major Concerns

  1. Both in the abstract and in the body of the review body (section 4 heading), the author states that, “the molecular basis of ESCRT recruitment to these sites (plasma membrane for HIV and inner nuclear membrane for HSV-1) is conserved,” suggesting that there is a topological trigger from hexamer formation which recruits ESCRTs. Yet, there is no obvious or confirmed L-domain in UL34. Moreover, unlike for HIV, no direct link has yet been found between NEC hexamer formation and the recruitment of ESCRTs. Therefore, there is yet any evidence showing the trigger for ESCRT recruitment in these two systems is conserved or the same.

  1. The NEC from HSV-1 and PRV can mediate scission in vitro suggesting that ESCRTs are not required for scission (Bigalke NComms 2014, Lorenz JBC 2015). How does the author reconcile this finding with the proposed ALIX + NEC mechanism?

  1. The author focuses on the most recent study from Arii et al., which showed that nuclear egress was reduced in the presence of the Vps4 dominant-negative mutant in HeLa cells. However, the dominant-negative mutant of Vps4 has no effect on the nuclear egress in HEK293 cells (Crump J. Virol., 2007). This discrepancy points at a possibly cell-specific role of ESCRT III in HSV nuclear egress, which should be acknowledged and discussed.

  1. Figure 1 is difficult to interpret and must be revised. There are two colored hexamers in Fig. 1A, but the legend implies that the hexamers in the figures represent capsids. If this is the case, then the figure suggests that multiple capsids (navy blue) are within one bud, which is incorrect. Also, the purple hexamer at the bottom is much smaller than the labeled 50-nm opening and if this is a viral capsid, it should be much bigger. If the navy-blue capsids are meant to represent NEC, that should be labeled. Additionally, the salmon and beige coloring in 1B is difficult to distinguish.

  1. The paragraph on page 4 “The formation of the Gag lattice is critical for the proper function of ESCRT in virus release,” is confusing to the reader. It is unclear which protein is recruited at what point and how the listed proteins interact/function together. The author claims that Gag oligomerization is “…critical for proper function of ESCRT in virus release,” yet it is unclear how this relates to the disruption of the ESCRT-I helical interface. How does ESCRT-I helical interface relate to Gag oligomerization and recruitment of ESCRT proteins?

A bigger, yet related concern is the general lack of explanation of how the ESCRT proteins work, especially, their role in scission. The review would greatly benefit from a description of the ESCRT function in uninfected cells and during HIV-mediated budding.

  1. In section 4, the author speculates that truncated lamin A, causing progeria, may result in aberrant binding of genomic DNA, which leads to ALIX recruitment due to the potential formation of, “equivalent membrane associated oligomers that mimic viral hexagonal lattices.” However, in the following paragraph, the author states that in Drosophila, large RNPs assemble, “as granules as opposed to highly organized lattices like that seen by HSV NEC or HIV Gag.” Given the statement about RNPs, would it not be more likely that the speculated Lamin A and genomic DNA interaction results in aggregates and not an oligomeric lattice? In the absence of any experimental evidence, the claim about lamin A comes across as an overinterpretation.

  1. “The NEC of HSV-1 forms Gag-like assemblies that recruit ALIX…” is an overintepretation considering that HSV-1 NEC and HIV Gag form distinct hexagonal lattices. More importantly, there is no evidence that the oligomerization of the NEC recruits ESCRTs.

Additional Comments

  1. There are many instances where a scientific claim is made, yet the reference does not come until a few sentences later. Citations should be inserted immediately after the statement is made. Further, incorrect references are used for some of the claims. For instance, the sentence with, “additionally, the UL31-UL34 heterodimer forms a lattice…” should cite Bigalke et al., 2014 and not Draganova et al., 2020. Reference to oligomeric interface mutants should include Bigalke 2015 in addition to Arii 2019. Reference 14 should be changed to Bigalke 2014 (HSV-1 NEC lattice in liposomes) and Hagen 2015 (NEC lattice in uninfected cells expressing PRV NEC).

  1. Panel A of a figure should be referred to before panel B.

  1. The author states, “….there are notable cellular binding partners that also promote primary envelopments…”, so it would be helpful to list a few examples and cite the sources.

  1. Figure 3 does not require A, B, etc. labeling because there is only one figure. It is unclear what structural elements represent experimentally determined structures vs. predictions. Different colors would be helpful here. A C-terminal alpha helix missing from HSV-1 UL34 structure but present in PRV, HCMV, MCMV, EBV should be modelled based on the homolog structures rather than the secondary structure prediction. The model does not include the N-terminal regions of UL31 or UL34.

  1. Numbering for the individual sections is incorrect for Conclusion, and #5 is missing.

  1. The sentence, “For instance, at the plasma membrane, ESCRT recruitment is essential for HIV assembly and budding, while it is indispensable for the release of HSV,” is confusing. Is the author trying to say that HIV and HSV are different or the same?

  1. In the introduction, the author refers to L domains as “Late domain” or “late budding domain”. This should be standardized.

  1. “African green” should read “African green monkey”.

  1. At the end of the first paragraph, the V domain of ALIX should be introduced.

  1. The paragraph starting with, “Flaviviruses on the other hand…” seems out of place. Perhaps it would be better placed near the end of the review highlighting the fact that the viral recruitment of ESCRTs has no clear mechanism. Further, replace “vesicles packets” with “vesicle packets”.

  1. In introduction section, replace “capsid core” with “capsid”. The text discussing figure 2 and the figure 2 legend should reflect the fact that the NEC lattice structure was reconstructed by fitting the NEC crystal structure into a 3D average of the NEC coat, which is from PRV and not HSV-1 (Hagen et al., Cell, 2015).

  1. The EMDB IDs listed in figure 2 are the PDB IDs of the crystal structures.

  1. In the first Introduction paragraph, HIV-1 is defined twice, as is the sentence following it introducing the L-domains.

Author Response

Major Concerns

  1. Both in the abstract and in the body of the review body (section 4 heading), the author states that, “the molecular basis of ESCRT recruitment to these sites (plasma membrane for HIV and inner nuclear membrane for HSV-1) is conserved,” suggesting that there is a topological trigger from hexamer formation which recruits ESCRTs. Yet, there is no obvious or confirmed L-domain in UL34. Moreover, unlike for HIV, no direct link has yet been found between NEC hexamer formation and the recruitment of ESCRTs. Therefore, there is yet any evidence showing the trigger for ESCRT recruitment in these two systems is conserved or the same.

 This line (16) of the abstract has be amended and now reads: “For instance, at the plasma membrane, ESCRT recruitment is essential for HIV assembly and budding, while it is dispensable for the release of HSV.” The author apologizes for this incorrect statement that was inaccurate in the previous manuscript submission.

  1. The NEC from HSV-1 and PRV can mediate scission in vitro suggesting that ESCRTs are not required for scission (Bigalke NComms 2014, Lorenz JBC 2015). How does the author reconcile this finding with the proposed ALIX + NEC mechanism?

 HIV Gag has also been shown to vesiculate membranes in vitro (see line 133-136). Line 192 was added to section 3 which includes the reviewer’s specified reference while drawing a comparison to HIV. Although fundamentally interesting, the fact that the NEC is capable of vesiculation in vitro, does not override the biochemical data that suggests that virus nuclear egress is sensitive to ALIX depletion and VPS4 mutagenesis.

  1. The author focuses on the most recent study from Arii et al., which showed that nuclear egress was reduced in the presence of the Vps4 dominant-negative mutant in HeLa cells. However, the dominant-negative mutant of Vps4 has no effect on the nuclear egress in HEK293 cells (Crump J. Virol., 2007). This discrepancy points at a possibly cell-specific role of ESCRT III in HSV nuclear egress, which should be acknowledged and discussed.

 The reviewer points out an important reference regarding the ability of the NEC to mediate budding independently of the ESCRTs. This reference has been included in the text and is discussed on lines 202-206.

  1. Figure 1 is difficult to interpret and must be revised. There are two colored hexamers in Fig. 1A, but the legend implies that the hexamers in the figures represent capsids. If this is the case, then the figure suggests that multiple capsids (navy blue) are within one bud, which is incorrect. Also, the purple hexamer at the bottom is much smaller than the labeled 50-nm opening and if this is a viral capsid, it should be much bigger. If the navy-blue capsids are meant to represent NEC, that should be labeled. Additionally, the salmon and beige coloring in 1B is difficult to distinguish.

 The colors in Figure 1 have been updated for clarity that there is a singular capsid separated from the membrane-associated NEC. Line 104 has been updated to the following: “A single virus capsid is shown as a collection of red hexamers and the NEC is shown as membrane-associated sticks (black).”

  1. The paragraph on page 4 “The formation of the Gag lattice is critical for the proper function of ESCRT in virus release,” is confusing to the reader. It is unclear which protein is recruited at what point and how the listed proteins interact/function together. The author claims that Gag oligomerization is “…critical for proper function of ESCRT in virus release,” yet it is unclear how this relates to the disruption of the ESCRT-I helical interface. How does ESCRT-I helical interface relate to Gag oligomerization and recruitment of ESCRT proteins?

A bigger, yet related concern is the general lack of explanation of how the ESCRT proteins work, especially, their role in scission. The review would greatly benefit from a description of the ESCRT function in uninfected cells and during HIV-mediated budding.

Lines 30-34 have been added to describe the ubiquity of ESCRTs in the cell.

Line 69 has been added which describes the membrane scaffolding role of ESCRTs.

Lines 125-130 describe the process of membrane scission via the ESCRT pathway, but line 128 has been amended to include the following: “ESCRT-I then assembles into a ring that templates the oligomerization of downstream ESCRT-III and the membrane scission machinery which abscises membranes by forming filaments within membrane necks.”

 Line 137 has been amended to the following for clarity: “Retroviral budding is a geometry-dependent process where the formation of the Gag lattice is critical for the proper recruitment and function of ESCRT in virus release.”

  1. In section 4, the author speculates that truncated lamin A, causing progeria, may result in aberrant binding of genomic DNA, which leads to ALIX recruitment due to the potential formation of, “equivalent membrane associated oligomers that mimic viral hexagonal lattices.” However, in the following paragraph, the author states that in Drosophila, large RNPs assemble, “as granules as opposed to highly organized lattices like that seen by HSV NEC or HIV Gag.” Given the statement about RNPs, would it not be more likely that the speculated Lamin A and genomic DNA interaction results in aggregates and not an oligomeric lattice? In the absence of any experimental evidence, the claim about lamin A comes across as an overinterpretation.

 Line 306 has been amended to the following: “For this to occur, DNA associated-defective  progerin intermediate filaments would have to adopt equivalent membrane associated oligomers that topologically mimic viral hexagonal lattices, albeit through a different yet mechanistically equivalent assembly that triggers ESCRT recruitment.” The goal of this statement was to highlight the parallels between progerin-mediated ALIX recruitment and HSV NEC-mediated ALIX recruitment and suggest not that progerin forms a hexagonal lattice, but rather that the membrane deformity caused by a progerin-genomic DNA assembly is a topologically equivalent RNP-induced membrane deformity to the HSV NEC.

  1. “The NEC of HSV-1 forms Gag-like assemblies that recruit ALIX…” is an overintepretation considering that HSV-1 NEC and HIV Gag form distinct hexagonal lattices. More importantly, there is no evidence that the oligomerization of the NEC recruits ESCRTs.

Although there is currently no direct evidence that ALIX recruitment to the NEC is triggered by the formation of the HSV NEC hexagonal lattice itself, the goal of this review was to draw on biochemical data from retroviruses to aid in the understanding of lesser known mechanisms like in the assembly and scission of the membrane bound HSV NEC. It certainly is intriguing that as depicted in Figure 2, that HIV Gag and HSV NEC form lattices with similar dimensions. Additionally, depletion of ALIX and inactivation of VPS4 display analogous defects in nuclear egress as they do on retroviral budding at the plasma membrane, suggesting that the ESCRTs are operating through a similar mechanism. The title of section 3, line 185,  has been amended to the following: “The NEC of HSV-1 forms Gag-like assemblies that are required for efficient nuclear egress and may trigger ALIX recruitment,” to address the reviewer’s points.

Additional Comments

  1. There are many instances where a scientific claim is made, yet the reference does not come until a few sentences later. Citations should be inserted immediately after the statement is made. Further, incorrect references are used for some of the claims. For instance, the sentence with, “additionally, the UL31-UL34 heterodimer forms a lattice…” should cite Bigalke et al., 2014 and not Draganova et al., 2020. Reference to oligomeric interface mutants should include Bigalke 2015 in addition to Arii 2019. Reference 14 should be changed to Bigalke 2014 (HSV-1 NEC lattice in liposomes) and Hagen 2015 (NEC lattice in uninfected cells expressing PRV NEC).

 The following references have been updated according to the reviewer’s suggestions.

  1. Panel A of a figure should be referred to before panel B.

 Panel a is now mentioned on line 80 before panel B.

  1. The author states, “….there are notable cellular binding partners that also promote primary envelopments…”, so it would be helpful to list a few examples and cite the sources.

 This line (201) has been amended to the following: “Additionally, there are notable cellular binding partners that also promote herpesvirus primary envelopment including the WD repeat-containing protein 5 (WDR5) and the ESCRT-associated protein ALIX.” This includes the respective references.

  1. Figure 3 does not require A, B, etc. labeling because there is only one figure. It is unclear what structural elements represent experimentally determined structures vs. predictions. Different colors would be helpful here. A C-terminal alpha helix missing from HSV-1 UL34 structure but present in PRV, HCMV, MCMV, EBV should be modelled based on the homolog structures rather than the secondary structure prediction. The model does not include the N-terminal regions of UL31 or UL34.

 Figure 3 is now a homology model of both HSV-1 UL31-UL34 based on threading of the PRV template (4z3u) and the figure legend and lines 230-235 have been updated accordingly.

  1. Numbering for the individual sections is incorrect for Conclusion, and #5 is missing.

 The numbering has been corrected for all sections and the conclusion is now part 5.

  1. The sentence, “For instance, at the plasma membrane, ESCRT recruitment is essential for HIV assembly and budding, while it is indispensable for the release of HSV,” is confusing. Is the author trying to say that HIV and HSV are different or the same?

 This was an incorrect statement regrettably overlooked by the author. This has been corrected on line 16 to “dispensable.”

  1. In the introduction, the author refers to L domains as “Late domain” or “late budding domain”. This should be standardized.

This has been standardized on line 28 and again on line 38 in agreement with the reviewer’s suggestion.

  1. “African green” should read “African green monkey”.

 The text on line 49 now reads “African green monkey.”

  1. At the end of the first paragraph, the V domain of ALIX should be introduced.

 The V domain of ALIX is introduced on line 50 with the following: “ALIX interactions with LYPXnL late domains are mediated by the central V domain of ALIX that has a hydrophobic cavity capable of accepting and stabilizing a tyrosine residue from an LYPXnL-containing binding partner. In contrast, ALIX recruitment by SIVs with atypical motifs reveals that ALIX can be recruited more simply to a partner protein with a tyrosine residue with the proper flexibility to embed within the V domain of ALIX, inde-pendently of the immediate flanking amino acid residues and the entire canonical LYPXnL motif.”

  1. The paragraph starting with, “Flaviviruses on the other hand…” seems out of place. Perhaps it would be better placed near the end of the review highlighting the fact that the viral recruitment of ESCRTs has no clear mechanism. Further, replace “vesicles packets” with “vesicle packets”.

 Line 58 now reads: “Flaviviruses, another class of RNA viruses that recruit ESCRT proteins,” to assist in the flow of the manuscript in agreement with the reviewer’s suggestion. Additionally, line 67 now reads “vesicle packets.”

  1. In introduction section, replace “capsid core” with “capsid”. The text discussing figure 2 and the figure 2 legend should reflect the fact that the NEC lattice structure was reconstructed by fitting the NEC crystal structure into a 3D average of the NEC coat, which is from PRV and not HSV-1 (Hagen et al., Cell, 2015).

 The figure legend of Figure 2 has been amended on lines 182-184 and includes the reviewer’s helpful suggestions for clarification and reference.

  1. The EMDB IDs listed in figure 2 are the PDB IDs of the crystal structures.

 Figure 2 now has the correct EMD accession numbers

  1. In the first Introduction paragraph, HIV-1 is defined twice, as is the sentence following it introducing the L-domains.

HIV-1 is now introduced just once on line 27.

Round 2

Reviewer 1 Report

Manuscript: When in need of an ESCRT: The structure of virus assembly sites reveal mechanistic parallels between nuclear egress and retroviral budding. (Version2)

Author: Rose, K.

Journal:  Viruses

The revision of the initial submission of this review is substantially improved and much easier to follow the author’s arguments. I have only minor concerns and these are limited to unconventional use of English.

Line 170 “geometric favorability”. I believe this statement could be improved by employing more conventional terminology. This would help audience of readers for which English is not primary language.

Line 208 “In opposition” maybe “In contrast”

Line 273 “evolutionary conclusion” I am not an evolutionary biologists but not sure I have heard of an evolutionary conclusion unless it is extinction.

Line 296 c-terminal change to C-terminal

Otherwise the author has answered my previous queries.

Reviewer 2 Report

The criticisms have been addressed only in a perfunctory manner, without a thoughtful consideration of the reviewers’ concerns. A few prominent examples of the missed opportunities to address the prior comments are listed below. These are by no means exhaustive, however, and the author is urged to go over all of the prior comments as well.

  1. The author still has the phrase in the abstract “... the molecular basis of ESCRT recruitment to these sites is conserved” (lines 18 and 19), even though it says it has been corrected in the author response to criticisms.

  1. Figure 1 is still unclear. Given that the NEC forms a hexagonal coat, the use of multiple red hexagons within the perinuclear space suggests this is the NEC coat, rather than one capsid (despite the legend description). There are plenty of examples of 2D representations of icosahedral capsids in the literature that the author could use for inspiration. Additionally, showing Vps4 as a hexagon is confusing.

  1. While some additional detail about ESCRTs were added to the introduction, a general description of the ESCRTs is still lacking. For example, ESCRT-I and ESCRT-III are both listed in the paragraph starting on line 140, yet the differences between the two have not yet been made clear to the read. Equally mirky is the difference between ALIX and TSG101 or how they fit into the HIV-1 Gag story. While this may be obvious to an ESCRT expert, it will be confusing to the broad readership of the Viruses.

  1. The hypothesis that the truncated lamin A in progeria may form a hexagonal lattice is overstated given the information provided in the following paragraph. The author needs to take a step back and propose either option (lattice or granules) considering the lack of evidence to support either.

  1. Figure 3 still looks odd. This could be, in part, due to the tilted as opposed to the vertical orientation of the NEC relative to the membrane. It is also unclear why the author made a homology model of the HSV-1 NEC when the crystal structure is available. If the presented model combines the structure and the homology model, this should be made clear by using colors.

  1. There are still many instances where scientific findings are stated and the citation is multiple sentences away.